# Brief Communication: Stay local or go global? On the construction of plausible counterfactual scenarios to assess flash flood hazards

Paul Voit[1] and Maik Heistermann[1]

[1]Institute for Environmental Sciences and Geography, University of Potsdam, Potsdam, Germany

**Correspondence:** Paul Voit (voit@uni-potsdam.de)

**Abstract.** Spatial counterfactuals are gaining attention to address the lack of robust flood frequency analysis in small catchments. However, the credibility of counterfactual scenarios decreases with the distance rain fields are transposed across space. We limit that distance by a local counterfactual search design, and compare the corresponding scenarios to recently published results from long-distance transpositions. We then put all scenarios in context with 200-year return levels, and with flood peaks simulated for the June 2024 flood event in southern Germany. We conclude that local counterfactuals scenarios are transparent and credible, and could complement the anticipation of low probability events.

## 1 Introduction

A flash flood is defined as "a localised flood with very high volumes of fast-flowing water, often carrying large debris, that rises very quickly, with an immediate threat to life" (Cave et al., 2009). These floods are among the most impactful natural disasters worldwide regarding damage and human casualties. Our ability to observe flash floods is fundamentally limited by their small spatio-temporal scale: for flash-flood prone catchments, stream gauges are scarce, or, if they exist, often destroyed by the actual event. Rain gauge networks or spaceborne remote-sensing products are, in turn, too sparse or too coarse, respectively, to capture the flood-triggering convective precipitation features.

Disaster risk management is typically based on local observations of the past, using the formalism of flood frequency analysis (FFA). However, the local rarity and the lack of long-term observational records, especially for small basins, challenges conventional FFA. Furthermore, FFA is based on the assumption, that the (extreme) events are independent and identically distributed, which is questionable under climate change. In essence, the recurrence of so-called "unprecedented" events (such as the ones in Braunsbach (2016) and Ahrtal (2021) in Germany, or Marche (2022), in Italy) demonstrates the difficulties that arise from conventional FFA in a risk management context.

Counterfactual thinking can help to address these challenges by creating different, but plausible, scenarios of how an event could have unfolded (Woo, 2019). Scenarios with a worse outcome than that of actual event ("downward counterfactuals") can provide valuable insights for disaster risk management and can support preparedness. In the context of flood hazard assessment, one option for counterfactual scenario design is to spatially transpose the location of a heavy precipitation event (HPE) in order to assess the impact that it could have effectuated elsewhere. Recently, this approach has attracted increasing attention in the European flood research community (e.g., Montanari et al., 2023; Merz et al., 2024; Voit and Heistermann, 2024; Vorogushyn

et al., 2024). Yet, it appears that these studies did not account for a substantial body of prior research, specifically in the United States, that is largely centered around the terms of probable maximum precipitation (PMP), probable maximum flood (PMF), and stochastic storm transposition (SST). As pointed out by one of the referees of this manuscript, these terms stand for about a century-long record of research and development that was comprehensively documented and reflected, e.g., by Hansen (1987)

and Fontaine and Potter (1989) and, about 40 years later, by Wright et al. (2020). The common denominator of these studies is the aim to anticipate, for any catchment of interest (CoI), physically plausible extreme rainfall scenarios by searching for previous records of extreme rain storms not only in the CoI itself, but in some neighbourhood region which is considered as "meteorologically homogeneous". The spatial "transposition" of the major storms towards the CoI is one component of PMP estimation, others being physically-based moisture maximisation and the use of envelope curves. PMFs can then be

obtained from PMP estimates by means of rainfall-runoff models. While the PMP/PMF approach does not yield exceedance probabilities, the idea of SST is to include the concept of storm transposition in a more rigorous statistical framework for flood frequency analysis: as the name suggests, the defining feature of SST is the random (stochastic) transposition of major storms from a search neighbourhood over a CoI. With the advancement of radar-based precipitation estimation, both PMP and SST were confronted with new opportunities to represent rainfall characteristics in space and time (Wright et al., 2014).

Despite the the evidently large body of literature around the concept of spatial counterfactuals or storm transposition, the key question remains about the adequate size of the transposition domain. With increasing distance, the assumption of "meteorological homogeneity" might become invalid, leading to a loss of credibility with regard to the resulting counterfactual scenarios. The definition of "meteorological homogeneity", however, remains elusive, specifically in the context of exceptional extreme events, although attempts were made recently towards a more formal definition that goes beyond a simple neighbor-

hood window (see Zhou et al., 2019, as an example).

Yet, the inherent trade-off between "credibility" and "finding the probable maximum" or the "worst case" (or, even, as Montanari et al. (2023) put it, the "impossible flood") will be difficult to resolve. In this paper, we hence follow a different approach in which we explore the sensitivity of simulated flood peak estimates on two very disparate assumptions on the size of the transposition domain which, for the sake of simplicity, we will refer to as "global" and "local" counterfactuals:

– **Global counterfactuals**: Recently, Voit and Heistermann (2024) identified the 10 most extreme precipitation events that had occurred over Germany between 2001 and 2022. By systematically transposing these events all across Germany, they created a total of 230,000 counterfactual precipitation scenarios, resulting in 829 million simulations of counterfactual flood peaks. They found that, on average, the counterfactual peaks exceeded the maximum original peak (between 2001 and 2022) by a factor of 5.3. While Voit and Heistermann (2024) also neglected to refer to previous research in the

field of PMP, PMF and SST, the scope of their simulation experiment, with a comprehensive transposition of events at the national-scale (Germany), was still unique (and also raised the question whether such long transposition distances have any credibility). We will, in this study, refer to such a large-scale transposition across the full spatial domain of the national radar-composite as "global counterfactuals".

– Alternatively, we suggest **local counterfactuals** as a more conservative approach: for each catchment in Germany, we select the most extreme rainfall event between 2001 and 2022 that occurred in a 20 km buffer around a catchment, and then simulate the runoff response that this rainfall would have caused in that catchment of interest.

For each catchment, we then compare the maximum peak discharge obtained from these counterfactual designs, local and global, to the corresponding 50 and 200-year return levels.

We will also briefly address a recent flood event that affected large parts of southern Germany in early June 2024 (Mohr et al., 2024). In the context of this event, there were various reports of flood peaks that exceeded a level of "a flood of low probability" (according to the EU flood directive) which, in Germany, is typically referred to as $HQ_{extreme}$ flood, and associated with a return period of 200 years. In an exemplary case study, we investigate how the simulated flood peaks for this event compare to the 200-year return level and the local counterfactual flood peaks, and discuss potential implications for flood risk management.

## 2 Data and methods

Large parts of the data and methods applied for the present study were documented in detail in Voit and Heistermann (2024). Hence, we only briefly recap the data, the hydrological model, and the design of the global counterfactual scenarios, and extend this by the documentation of the flood frequency analysis and the selection of the local counterfactuals.

### 2.1 Precipitation Data

We used the radar climatology product (RADKLIM v2017.002) for the years 2001-2022, for the computation of global and local counterfactuals as well as for the continuous runoff modelling for Germany. The product is provided by Germany's national meteorological service (Deutscher Wetterdienst; DWD hereafter). RADKLIM is a reprocessed (Lengfeld et al., 2019) version of the DWD's operational radar-based quantitative precipitation estimation product (RADOLAN, see Winterrath et al., 2012). The data set has a spatial resolution of 1 x 1 km and a temporal resolution of one hour and is openly accessible on the DWD open data server (Winterrath et al., 2018). To model the flood peaks during the flooding in the Danube, Main and Neckar catchment in June 2024, we used the operational RADOLAN product instead, because RADKLIM is only updated on an annual cycle.

### 2.2 Digital elevation model

For the catchment delineation and the runoff analysis we used the EU-DEM. This DEM has a resolution of 25 m and is a combination of SRTM (Shuttle Radar Topography Mission) and ASTER GDEM (Advanced Spaceborne Thermal Emission and Reflection Radiometer Global Digital Elevation Model). The data set is available at the Copernicus Land Monitoring service (European Commission, 2016).

### 2.3 Land cover and soil data

As a basis for the SCS-CN method (U.S. Department of Agriculture-Soil Conservation Service, 1972) to estimate the effective
precipitation, we used the CORINE CLC5-2018 (BKG, 2018) for land cover and the "BUEK 200" (national soil survey at a
scale of 1:200,000; BGR, 2018) for soil data.

### 2.4 Hydrological model

We specifically tailored the hydrological model to represent flash flood events in small- to medium-sized basins. A comprehen-
sive model description can be found in Voit and Heistermann (2024). During flash flood events, surface runoff is the dominant
process (Marchi et al., 2010; Grimaldi et al., 2010) while evaporation and groundwater dynamics are negligible. For this reason
the model consists of only two modules. First, the effective rainfall is estimated using the SCS-CN method (U.S. Department
of Agriculture-Soil Conservation Service, 1972). The SCS-CN method is widely used in flash flood modelling while more
advanced modelling approaches are difficult to parameterize specifically in small catchments. Secondly, the geomorphological
instantaneous unit hydrograph (GIUH), as derived from the DEM, is used to represent the concentration of quick runoff (i.e. of
the effective rainfall). The light-weight design of the model allows for the computation of a large number of counterfactuals.
Because the model does not include channel mechanics and hydro-engineering measures, we restrict our analysis to catchments
with an area of less than $750\,\mathrm{km^2}$. The remaining 19809 sub-catchments have an average size of $15\,\mathrm{km^2}$. To make the modelled
peaks for the different subbasin sizes comparable, we use the unit peak discharge (UPD in $\mathrm{m^3/s/(km^2)^{0.6}}$, see Castellarin,
2007). The UPD is the ratio between the runoff peak (in $\mathrm{m^3/s}$) and the reduced catchment area (in $(\mathrm{km^2})^{0.6}$, as in Gaume
et al., 2008).

### 2.5 Flood frequency analysis

We model the quick runoff for each subbasin and for the whole length of the RADKLIM dataset (2001-2022), select the yearly
maxima of the UPD, fit a GEV-distribution for each subbasin, and estimate the 200- and 50-year return levels of UPD. We will
use both return levels as references for our analysis. Given the length of our yearly maxima series (2001-2022), we consider the
estimation of the 50-year return level as reasonably robust, while the 200-year return level will obviously be highly uncertain.

### 2.6 Development of counterfactual scenarios

As outlined in Sect. 1, we compare peak discharge from global and local counterfactual scenarios. The global counterfactuals
are the same as presented in Voit and Heistermann (2024): we selected the ten most extreme heavy precipitation events from
2001 to 2022, transposed them all across Germany and simulated the corresponding peak discharge for each subbasin in
Germany.

To provide more plausible and credible scenarios, we suggest a new approach which we refer to as "local counterfactuals".
It is based on the selection of heavy precipitation events from a neighbourhood around any *catchment of interest* (CoI, which is

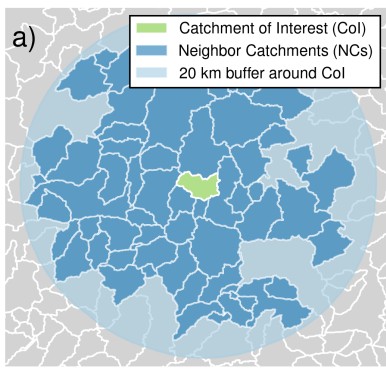 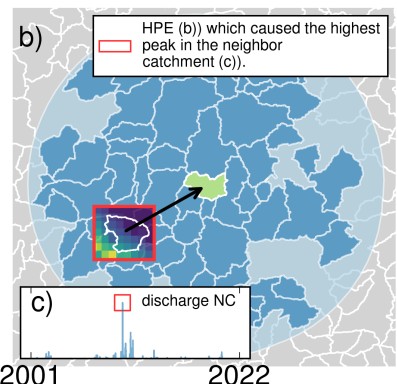 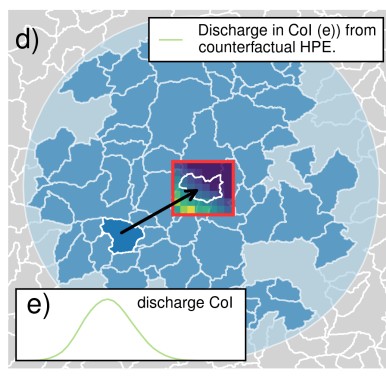

**Figure 1.** Development of local counterfactuals: a) Catchment of Interest (CoI, green) and its neighbor catchments (NCs, dark blue) in a 20 km neighborhood (light blue). b) Selecting the event which caused the highest runoff peak (c) in the NC (red box). d) Transposing the rainfall from the NC to the CoI and modelling the resulting runoff (e)). This procedure is repeated for each NC.

the catchment to which the counterfactual scenarios should be applied). As CoI, we consider each catchment in Germany that is smaller than 750 km$^2$, and apply the following steps (see also Fig. 1 for illustration):

1. For each CoI, we select all catchments which are fully contained in a 20 km buffer around the CoI. We refer to these as *neighbour catchments* (NCs, see Fig. 1a). On average, each CoI has 89 NCs.

2. For each of these NCs, we model the quick runoff from 2001 until 2022 (Fig. 1b). We then identify the date of the maximum peak discharge during this period (Fig. 1c).

3. From RADKLIM, we extract the data for the rainfall event which caused the highest peak in the NC (Fig. 1b) and
transpose it from its original spatial position to the centroid of the CoI, thereby creating a spatial counterfactual (Fig. 1d). We ensure that the CoI and all its upstream catchments will be completely covered by the rainfall event, by adding a large buffer on each side of the RADKLIM slice (for better visualization we do not show the buffer in Fig. 1).

4. We model the surface runoff that this counterfactual rainfall event would cause in the CoI (Fig. 1e) and record the peak discharge. We repeat steps 3 and 4 for all NCs.

5. Finally, we pick the highest counterfactual peak across all NCs (including the CoI, if none of the counterfactual peaks were higher) and keep this value as the "local counterfactual peak discharge" for later analysis.

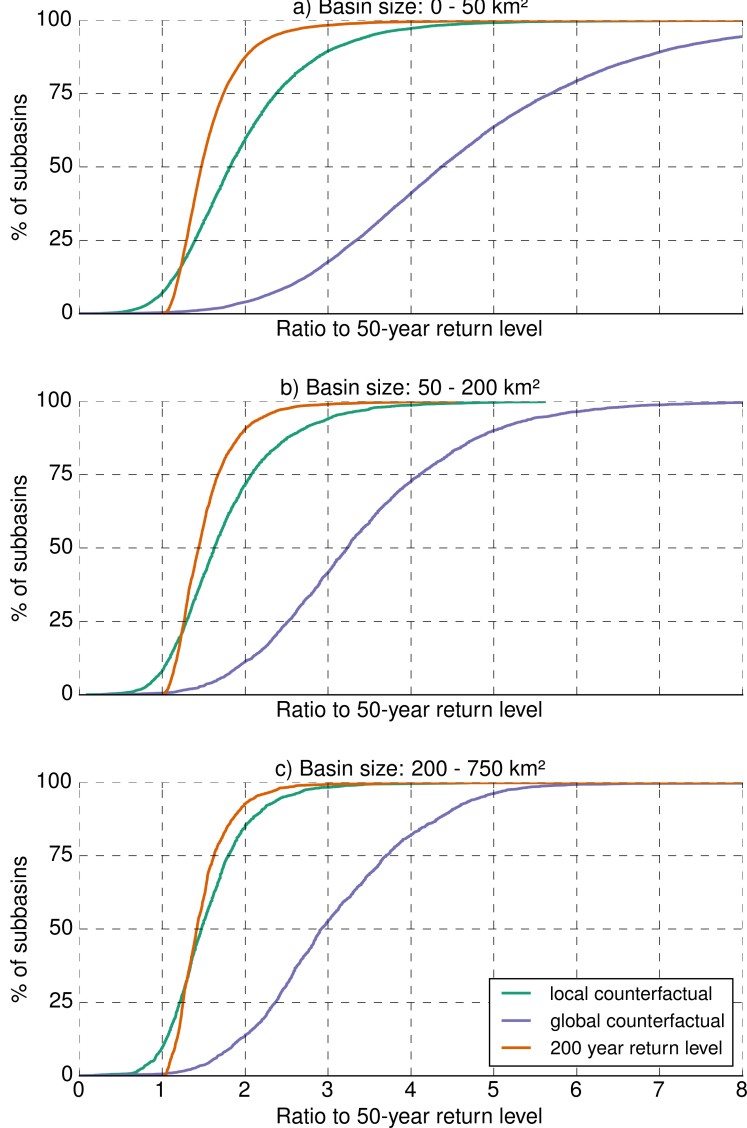

**Figure 2.** The cumulative density distributions show, for different subbasin sizes [a) $<50$ km$^2$, b) 50-200 km$^2$, c) 200-750 km$^2$], the ratio between three different discharge estimates and the 50-year return level: (1) the local counterfactual peak discharge (green), (2) the global counterfactual peak discharge (purple), (3) the 200-year return level (orange).

# 3   Results and discussion

## 3.1   Local versus global counterfactuals

For each basin, we compute the ratio between the global counterfactual unit peak discharge and the corresponding 50-year
return level. We do the same for the local counterfactual UPD. As an additional reference, we compute the ratio between the
200-year and the 50-year return level. Fig. 2 shows the cumulative distributions of the resulting ratios across three classes of
basin sizes. The global counterfactuals effectuate by far the highest peak discharge (i.e. ratio) at all spatial scales. While this
is unsurprising, the extent at which local counterfactuals and 200-year return levels are dwarfed by the global counterfactual
peaks remains impressive – and alarming. Thinking in terms of flood frequency analysis, these peaks seem beyond any notion
of a return period.

The curves for the local counterfactuals and the 200-year return level are much closer to each other. For increasing basin
sizes, the local counterfactual curves approach the 200-year return level curves (which are relatively stable across basin sizes),
until both are are nearly congruent for basin sizes larger than 200 km$^2$.

There might be different reasons behind this scale dependency. Small-scale convective heavy rainfall events tend to cause a
stronger runoff response in small catchments, but they are also more likely to closely *miss* a small catchment. We would hence
expect the local counterfactual search to be more efficient in the process of finding small-scale precipitation events in a CoI's
neighbourhood and displacing them right over that CoI to produce an exceptional flood response. Furthermore, we observe
a general leftward shift of all curves (including the global counterfactuals) with increasing basin size and increasing flood
magnitude (i.e. ratio). This could be explained by flood hydrographs becoming more attenuated with increasing catchment size
due to the spatio-temporal convolution of the rainfall input.

As a consequence, future studies could investigate how to adjust the local counterfactual search for the effects of scale.
For instance, we could select local counterfactuals for the CoI exclusively from similarly sized neighbour catchments. We
could then also explore a larger number of realizations when displacing the rainfall field over the CoI, in order to capture
constellations in which the spatio-temporal convolution maximizes the peak discharge. In this context, a scale-adjusted search
buffer around the CoI might also be justified. Generally, the choice of the buffer for the selection of neighbor catchments has a
strong influence on the outcome of the counterfactual study. We arbitrarily chose a 20 km buffer size. Further investigation is
needed, to decide until which buffer size counterfactuals from neighbor catchments are plausible and which other parameters
could be included in the selection process. The number of counterfactuals could be increased by not just transposing the HPE
which caused the highest runoff peak during 22 years, but using all the events which caused the yearly runoff maxima.

## 3.2   The 2024 summer flood in southern Germany

We now compare the local counterfactual peaks and the 200-year return levels to the peak discharge values which we simulated
for the recent flood event in southern Germany. The floods were caused by heavy precipitation from May 30 to June 4, 2024,
with most of the rainfall accumulating on May 30 and June 1. The event caused large damages specifically along the southern
tributaries of the Danube. For a detailed synopsis of the event we refer to (Mohr et al., 2024, in German). For our comparison,

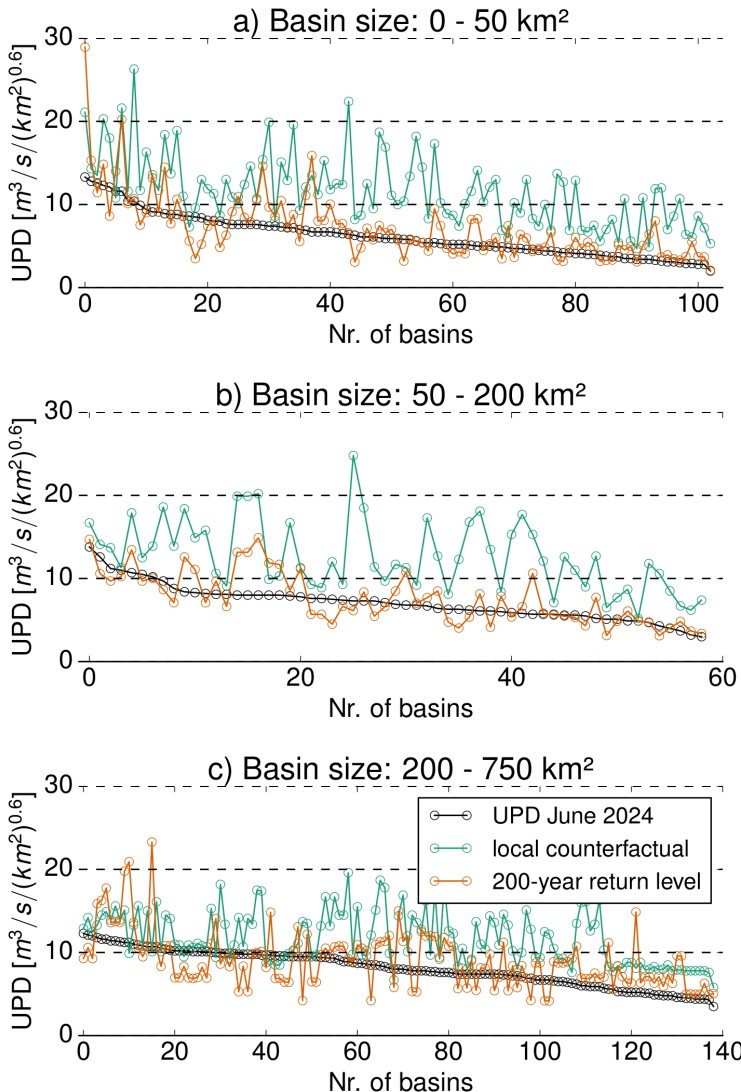

**Figure 3.** Case study of the recent heavy precipitation event from May 30 to June 4, 2024: the black lines show the simulated unit peak discharge (UPD) of the event for all subbasins within the Danube catchment with a return period > 20 years; for comparison, the green lines show the local counterfactual UPD and the orange lines the 200-year return level estimated from simulations between 2001 and 2022.

we select all subbasins of the German Danube basin for which the simulated UPD of the June 2024 event exceeded a 20-year return level (301 basins), and compare this UPD to the respective local counterfactual peaks and the 200-year return level. As Fig. 3 shows, the peak discharge during the June 2024 event exceeded the (simulated) 200-year return levels in 36% of the selected subbasins. In contrast, the local counterfactual peaks were exceeded in only 5 % of the subbasins. This effect is less pronounced for catchments which are larger than 200 km$^2$.

For this recent event, the concept of local counterfactuals could have helped to anticipate the flood levels. Of course, we need to acknowledge the large uncertainties associated with this case study, specifically with regard to the validity of the hydrological model and with regard to the estimation of the 200-year return level from just 22 years of data – which is, strictly speaking, off limits. Yet, our example merely demonstrates how local counterfactuals – which we consider as credible scenarios – could complement inherently uncertain estimates of return levels for low-probability floods.

## 4   Conclusions

Global counterfactuals effectuate peak discharge levels that are typically far beyond any reasonable notion of return periods. This holds even more for very small catchments. Unsurprisingly, local counterfactuals are much less extreme than global ones. They appear to be closer to the runoff response that would correspond to return periods of several hundreds of years. The larger the basin size, the more the runoff response of local counterfactuals approaches the estimated 200-year return level.

That way, local counterfactuals could be in the order of flood levels that are typically associated to what the European Union's "flood directive" (European Commission, Directorate-General for Environment, 2013, article 6.3a) refers to as "floods of low probability, or extreme event scenarios" which is generally interpreted as a flood with a return period that is much higher than 100 years. Many member states, including Germany, have set the corresponding return period of such "extreme event scenarios" to 200 years.

As the estimation of peak discharge for such long return periods is obviously and inherently limited in the face of short time series, local counterfactuals could complement return levels that are conventionally estimated from discharge gauge records. The approach is robust, plausible, transparent and straightforward to communicate: if a precipitation event could happen 20 kilometers from here, it could as well happen right on your doorstep – so better be prepared for the resulting flood (evidently, the actual transposition distance should be subject to further discussion).

Still, we do not suggest to abandon the concept of "global counterfactuals". While counterfactual scenarios loose credibility with increasing transposition distance, it is exactly this type of counterfactual search that could aid flood risk management to make the transition from "unprecedented and therefore unimaginable, unexpected and unprepared" to "unprecedented but anticipated". Future research should hence explore new ways, including atmospheric modelling, to assess how the plausibility of spatial counterfactual precipitation scenarios depends on transposition distance.

In their review paper on SST, Wright et al. (2020) already noted that "SST research has been generally confined to the United States and Australia". Apparently, there is a gap between the flood research communities in the US and Europe with regard to the concept of spatial transposition (one might be inclined to phrase this as the European research lagging behind). In any case, applications in Europe are rare (see Lompi et al., 2022, as an example), and many researchers and practitioners may not be fully aware of how the recent concept of "spatial counterfactuals" relates to the established ideas of storm transposition (this

had certainly applied to the authors of this study before they were enlightened by one of the referees). On a positive note, this paper could, hopefully, do its bit to close the aforementioned gap, raise awareness of previous work, unify research efforts, and support the momentum which the application of these concepts has recently experienced.

*Code and data availability.*

We published notebooks and code which demonstrate our hydrological model for a small, exemplary region (Altenahr
basin): the derivation of GIUHs from a digital elevation model, the extraction of rainfall data from and effective rainfall for the
subbasins from RADKLIM data and the modelling of quick runoff. The code is published at: https://doi.org/10.5281/zenodo.
10473424.

All data used in this study is accessible at the open data repository of the DWD: the RADKLIM_RW_2017.002 dataset is
available at https://opendata.dwd.de/climate_environment/CDC/grids_germany/hourly/radolan/reproc/2017_002, (Winterrath
et al., 2018); the EU-DEM is available at https://ec.europa.eu/eurostat/web/gisco/geodata/digital-elevation-model/eu-dem#
DD, (European Commission, 2016); the CLC5-2018 land cover data is available at https://gdz.bkg.bund.de/index.php/default/
open-data/corine-land-cover-5-ha-stand-2018-clc5-2018.html, (BKG, 2018). The soil data is available at https://www.bgr.
bund.de/DE/Themen/Boden/Informationsgrundlagen/Bodenkundliche_Karten_Datenbanken/BUEK200/buek200_node.html, (BGR,
2018) All data last accessed 27 June 2024.

*Author contributions.*   PV and MH conceptualized this study. PV developed the software and carried out the analysis; MH contributed to the
analysis. PV and MH wrote the manuscript.

*Competing interests.*   The contact author has declared that neither they nor their co-authors have any competing interests.

*Acknowledgements.*   We would like to thank the open source community without its software and data this study would have not been
possible. Some small parts of the text were improved in exchange with a language model (https://chat.openai.com/chat)

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
