# Peer review of "Brief Communication: Stay local or go global? On the construction of plausible counterfactual scenarios to assess flash flood hazards"

_Natural Hazards and Earth System Sciences, 2024_

## Author Comment (AC1)

**Interactive Discussion: Author Response to Referee #1**

**Brief Communication: Stay local or go global? On the construction of plausible counterfactual scenarios to assess flash flood hazards**

Paul Voit and Maik Heistermann

*NHESS Discussions,* `doi:10.5194/nhess-2024-119`
* * *
**RC:** *Reviewer Comment*,     AR: *Author Response*,     ☐ Manuscript text

Dear Referee,

we are grateful for the time and effort you invested in this review, and for your critical and constructive comments. We really appreciate the concise and comprehensive synopsis on both PMP and SST, and we hope (and are confident) to find a way to adequately represent this information in our manuscript.

Please find our point-by-point responses to your comments below. These should be considered as preliminary (part of the interactive discussion). The actual revision of the manuscript, including another comprehensive response letter also depends on another referee report and the subsequent editorial decision.

Thanks again for your efforts!

Kind regards,
Paul Voit and Maik Heistermann

**RC:** *In what I consider to be a failure of the literature review and peer review process, the present study and those previous ones neglect the century-long massive body of relevant research that in multiple respects is much more advanced what the authors present in terms of level of sophistication and importantly, real-world application. Namely, geographic transposition of storms to generate counterfactual flood scenarios has been a cornerstone of flood risk management in North America, Australia, and elsewhere (perhaps not in Europe) for about as long as flood risk management has been quantified. [...] Proper acknowledgement of the efforts of earlier researchers is certainly needed. Unfortunately, the scope for doing so in the case of the two already-published studies is limited. In order for the present study to be published in NHESS or any other publication, the authors would need to adequately review and cite existing research and practice, as well as carefully place their own work within this much broader 100+-year long body of work.*

**AR:** We were honestly astounded when receiving your comments, and have to admit that we did not come across these concepts during our previous research; neither had the topic of "stochastic storm transposition" (and the fact that this is a common concept in the USA and Australia) come up during numerous presentations at conferences and workshops, or discussions with fellow researchers and previous reviewers.

We certainly accept the responsibility for this, and it is kind of discouraging that, despite modern search engines, a mere difference in terminology ("storm transposition" versus "spatial counterfactuals") can lead to

overlooking significant aspects of previous research. Of course, such things have happened before, and our neglect maybe highlights a more general issue when scientific communities remain unaware of each other's work. All the more, we think that it is important to acknowledge the gap between research communities in Europe and the US, and to seek to close it. In fact, this paper could be an opportunity and a starting point to explicitly address this issue, and to contribute to unify research efforts.

In order to adequately (though still concisely) represent the previous research on PMP and SST in the context of our manuscript, we kindly ask the editor whether the format restrictions of the "Brief communication" can be slightly relaxed in order to use a bit more than 20 references. We would then provide a revised version of the manuscript subject to the comments of the second referee and the following editorial decision.

We suggest the following changes to the manuscript, all bearing in mind the important requirement of brevity in the context of a brief communication:

- in the introduction, we will briefly introduce the concepts of PMP and SST, based on the excellent references provided by the referee (particularly Wright et al., 2020) and put these into context with the recent studies on spatial counterfactuals in Germany.

- furthermore, we would like to pick up this issue in the conclusions, and to openly address the apparent gap between the US/Australia and namely Europe in order to provide a perspective on how to unify the efforts in research and specifically applications (or maybe, rather, to allow the European community to catch up on what has already been done in the US).

**RC:** *Generally: there's no reason why this approach needs to be restricted to flash floods.*

AR: This is true, and the recent studies of Merz et al. (2024) and Vorogushyn et al. (2024) have demonstrated this. However, the focus of our work (referring to Voit and Heistermann (2024)) is on the application of this concept to catchments that are prone to flash floods: the rare coincidence of extreme precipitation with basins that are capable of producing flash flood, together with the data scarcity in such catchments, especially calls for a spatial counterfactual (or PMP/SST) approach, as the lack of historical experience may result in low risk awareness.

The focus on flash floods requires specific considerations with regard to the temporal and spatial resolution of precipitation observations, the robustness, parsimoniousness and computational efficiency of the hydrological model, and the resolution (or incremental distance) at which precipitation fields are transposed across a catchment of interest in order to actually find counterfactual scenarios that effectuate maximum peak flows. Applying this at the national scale (here: Germany-wide) implies a very large number of counterfactual scenarios and hence large computational expense: e.g., for our previous study (Voit and Heistermann, 2024), we modelled close to a billion hydrographs, stemming from 23.000 counterfactual scenarios.

**RC:** *Given the major criticism above, it would be appropriate to adopt existing language (e.g., "transposing" instead of "shifting")*

AR: We agree and will change "shift" to "transpose". We will also use the term "transposition domain". After reading the references you provided we noticed that we, at least and by chance, also came up with the commonly used term "Catchment of interest".

**RC:** *L15: the meaning of "small-scale observational records" is unclear.*

AR: We changed it to:

> However, the local rarity and the lack of long-term observational records, especially for small basins, challenges conventional FFA.

**RC:** *Should the language be clarified to make clear that it is an observed heavy precipitation event that is transposed?*

 **AR:** We suggest to clarify this as follows:

> In the context of flood hazard assessment, one option for counterfactual scenario design is to spatially transpose the location of an observed heavy precipitation event in order to assess the impact that it could have effectuated elsewhere.

**RC:** *Figures 2 and 3: these figures did not render properly in the PDF I downloaded, using two different widely-used PDF viewers on a Mac. Given that, I can't properly assess these figures.*

 **AR:** We are sorry for this. We do not know why the figures did not render properly, and have not yet heard about any such issue, yet. In any case, we enclose the figures at the end of this response letter for your perusal, and hope that the issue can be resolved in the course of the revision process together with Copernicus' editorial office.

**References**

Merz, B., Nguyen, V. D., Guse, B., Han, L., Guan, X., Rakovec, O., Samaniego, L., Ahrens, B., and Vorogushyn, S.: Spatial counterfactuals to explore disastrous flooding, Environmental Research Letters, 10.5194/nhess-2024-11910.1088/1748-9326/ad22b9, 2024.

Voit, P. and Heistermann, M.: A downward counterfactual analysis of flash floods in Germany, Natural Hazards and Earth System Sciences Discussions, 2024, 1–23, 10.5194/nhess-2024-11910.5194/nhess-2023-224, 2024.

Vorogushyn, S., Han, L., Apel, H., Nguyen, V. D., Guse, B., Guan, X., Rakovec, O., Najafi, H., Samaniego, L., and Merz, B.: It could have been much worse: spatial counterfactuals of the July 2021 flood in the Ahr valley, Germany, Natural Hazards and Earth System Sciences Discussions, 2024, 1–39, 10.5194/nhess-2024-11910.5194/nhess-2024-97, 2024.

[Figure]

Figure 1: The cumulative density distributions show, for different subbasin sizes [a) < 50 km$^2$, b) 50-200 km$^2$, c) 200-750 km$^2$], the ratio between three different discharge estimates and the 50-year return level: (1) the local counterfactual peak discharge (green), (2) the global counterfactual peak discharge (purple), (3) the 200-year return level (orange).

[Figure]

Figure 2: Case study of the recent heavy precipitation event from May 30 to June 4, 2024: the black lines show the simulated unit peak discharge (UPD) of the event for all subbasins within the Danube catchment with a return period > 20 years; for comparison, the green lines show the local counterfactual UPD and the orange lines the 200-year return level estimated from simulations between 2001 and 2022.

---

## Author Response (AR1)

**Interactive Discussion: Author Response to Referee #1**

**A downward counterfactual analysis of flash floods in Germany**

Paul Voit and Maik Heistermann
*NHESS Discussions,* `doi:10.5194/nhess-2023-224`
* * *
**RC:** *Reviewer Comment*,     AR: *Author Response*,     ☐ Manuscript text

Dear Editor,

thank you for coordinating the peer review process and for agreeing to include more references than usual in the manuscript. We believe that the manuscript has improved substantially during the review process.

Please find our responses to your comments below as well as the responses to the review of the two referees.

Thanks again and kind regards,

Paul Voit and Maik Heistermann

**1.  Comments and responses to the Referee 1**

**RC:** *In what I consider to be a failure of the literature review and peer review process, the present study and those previous ones neglect the century-long massive body of relevant research that in multiple respects is much more advanced what the authors present in terms of level of sophistication and importantly, real-world application. Namely, geographic transposition of storms to generate counterfactual flood scenarios has been a cornerstone of flood risk management in North America, Australia, and elsewhere (perhaps not in Europe) for about as long as flood risk management has been quantified. [...] Proper acknowledgement of the efforts of earlier researchers is certainly needed. Unfortunately, the scope for doing so in the case of the two already-published studies is limited. In order for the present study to be published in NHESS or any other publication, the authors would need to adequately review and cite existing research and practice, as well as carefully place their own work within this much broader 100+-year long body of work.*

 AR:  We were honestly astounded when receiving your comments, and have to admit that we did not come across these concepts during our previous research; neither had the topic of "stochastic storm transposition" (and the fact that this is a common concept in the USA and Australia) come up during numerous presentations at conferences and workshops, or discussions with fellow researchers and previous reviewers.

We certainly accept the responsibility for this, and it is kind of discouraging that, despite modern search engines, a mere difference in terminology ("storm transposition" versus "spatial counterfactuals") can lead to overlooking significant aspects of previous research. Of course, such things have happened before, and our neglect maybe highlights a more general issue when scientific communities remain unaware of each other's

work. All the more, we think that it is important to acknowledge the gap between research communities in Europe and the US, and to seek to close it. In fact, this paper could be an opportunity and a starting point to explicitly address this issue, and to contribute to unify research efforts.

In order to adequately (though still concisely) represent the previous research on PMP and SST in the context of our manuscript, the editor kindly agreed to relax the format restrictions of the "Brief communication" in order to use a bit more than 20 references.

We suggest the following changes to the manuscript, all bearing in mind the important requirement of brevity in the context of a brief communication:

**Beginning of revised introduction after l. 22**

.... In the context of flood hazard assessment, one option for counterfactual scenario design is to spatially transpose the location of a heavy precipitation event (HPE) in order to assess the impact that it could have effectuated elsewhere.

Recently, this approach has attracted increasing attention in the European flood research community (e.g., Montanari et al., 2023; Merz et al., 2024; Voit and Heistermann, 2024; Vorogushyn et al., 2024). Yet, it appears that these studies did not account for a substantial body of prior research, specifically in the United States, that is largely centered around the terms of probable maximum precipitation (PMP), probable maximum flood (PMF), and stochastic storm transposition (SST). As pointed out by one of the referees of this manuscript, these terms stand for about a century-long record of research and development that was comprehensively documented and reflected, e.g., by Hansen (1987); Fontaine and Potter (1989) and, about 40 years later, by Wright et al. (2020). The common denominator of these studies is the aim to anticipate, for any catchment of interest (CoI), physically plausible extreme rainfall scenarios by searching for previous records of extreme rain storms not only in the CoI itself, but in some neighbourhood region which is considered as "meteorologically homogeneous". The spatial "transposition" of the major storms towards the CoI is one component of PMP estimation, others being physically-based moisture maximisation and the use of envelope curves. PMFs can then be obtained from PMP estimates by means of rainfall-runoff models. While the PMP/PMF approach does not yield exceedance probabilities, the idea of SST is to include the concept of storm transposition in a more rigorous statistical framework for flood frequency analysis: as the name suggests, the defining feature of SST is the random (stochastic) transposition of major storms from a search neighbourhood over a CoI. With the advancement of radar-based precipitation estimation, both PMP and SST were confronted with new opportunities to represent rainfall characteristics in space and time (Wright et al., 2014).

Despite the the evidently large body of literature around the concept of spatial counterfactuals or storm transposition, the key question remains about the adequate size of the transposition domain. With increasing distance, the assumption of "meteorological homogeneity" might become invalid, leading to a loss of credibility with regard to the resulting counterfactual scenarios. The definition of "meteorological homogeneity", however, remains elusive, specifically in the context of exceptional extreme events, although attempts were made recently towards a more formal definition that goes beyond a simple neighborhood window (see Zhou et al., 2019, as an example).

Yet, the inherent trade-off between "credibility" and "finding the probable maximum" or the "worst case" (or, even, as Montanari et al. (2023) put it, the "impossible flood") will be difficult to resolve. In this paper, we hence follow a different approach in which we explore the sensitivity of simulated flood peak estimates on two very disparate assumptions on the size of the transposition domain which, for the sake of simplicity, we will refer to as "global" and "local" counterfactuals:

- **Global counterfactuals**: Recently, Voit and Heistermann (2024) identified the 10 most extreme precipitation events that had occurred over Germany between 2001 and 2022. By systematically transposing these events all across Germany, they created a total of 230,000 counterfactual precipitation scenarios, resulting in 829 million simulations of counterfactual flood peaks. They found that, on average, the counterfactual peaks exceeded the maximum original peak (between 2001 and 2022) by a factor of 5.3. While Voit and Heistermann (2024) also neglected to refer to previous research in the field of PMP, PMF and SST, the scope of their simulation experiment, with a comprehensive transposition of events at the national-scale (Germany), was still unique (and also raised the question whether such long transposition distances have any credibility). We will, in this study, refer to such a large-scale transposition across the full spatial domain of the national radar-composite as "global counterfactuals".

- Alternatively, we suggest **local counterfactuals** as a more conservative approach: for each catchment in Germany, we select the most extreme rainfall event between 2001 and 2022 that occurred in a 20 km buffer around a catchment, and then simulate the runoff response that this rainfall would have caused in that catchment of interest.

- in the introduction, we will briefly introduce the concepts of PMP and SST, based on the excellent references provided by the referee (particularly Wright et al., 2020) and put these into context with the recent studies on spatial counterfactuals in Germany.

- furthermore, we would like to pick up this issue in the conclusions, and to openly address the apparent gap between the US/Australia and namely Europe in order to provide a perspective on how to unify the efforts in research and specifically applications (or maybe, rather, to allow the European community to catch up on what has already been done in the US).

**End of revised introduction after l. 22**

Furthermore, we would like to pick up this issue in the conclusions, and to openly address the apparent gap between the US/Australia and namely Europe in order to provide a perspective on how to unify the efforts in research and specifically applications (or maybe, rather, to allow the European community to catch up on what has already been done in the US). We changed the conclusion as follows by adding one final paragraph:

> In their review paper on SST, Wright et al. (2020) already noted that "SST research has been generally confined to the United States and Australia". Apparently, there is a gap between the flood research communities in the US and Europe with regard to the concept of spatial transposition (one might be inclined to phrase this as the European research lagging behind). In any case, applications in Europe are rare (see Lompi et al., 2022, as an example), and many researchers and practitioners may not be fully aware of how the recent concept of "spatial counterfactuals" relates to the established ideas of storm transposition (this had certainly applied to the authors of this study before they were enlightened by one of the referees). On a positive note, this paper could, hopefully, do its bit to close the aforementioned gap, raise awareness of previous work, unify research efforts, and support the momentum which the application of these concepts has recently experienced.

**RC:** *Generally: there's no reason why this approach needs to be restricted to flash floods.*

AR: This is true, and the recent studies of Merz et al. (2024) and Vorogushyn et al. (2024) have demonstrated this. However, the focus of our work (referring to Voit and Heistermann, 2024) is on the application of this

concept to catchments that are prone to flash floods: the rare coincidence of extreme precipitation with basins that are capable of producing flash flood, together with the data scarcity in such catchments, especially calls for a spatial counterfactual (or PMP/SST) approach, as the lack of historical experience may result in low risk awareness.

The focus on flash floods requires specific considerations with regard to the temporal and spatial resolution of precipitation observations, the robustness, parsimoniousness and computational efficiency of the hydrological model, and the resolution (or incremental distance) at which precipitation fields are transposed across a catchment of interest in order to actually find counterfactual scenarios that effectuate maximum peak flows. Applying this at the national scale (here: Germany-wide) implies a very large number of counterfactual scenarios and hence large computational expense: e.g., for our previous study (Voit and Heistermann, 2024), we modelled close to a billion hydrographs, stemming from 23.000 counterfactual scenarios.

**RC:**   *Given the major criticism above, it would be appropriate to adopt existing language (e.g., "transposing" instead of "shifting")*

AR:   We agree and will change "shift" to "transpose". We will also use the term "transposition domain". After reading the references you provided we noticed that we, at least and by chance, also came up with the commonly used term "Catchment of interest".

**RC:**   *L15: the meaning of "small-scale observational records" is unclear.*

AR:   We changed it to:

> However, the local rarity and the lack of long-term observational records, especially for small basins, challenges conventional FFA.

**RC:**   *Should the language be clarified to make clear that it is an observed heavy precipitation event that is transposed?*

AR:   We suggest to clarify this as follows:

> In the context of flood hazard assessment, one option for counterfactual scenario design is to spatially transpose the location of an observed heavy precipitation event in order to assess the impact that it could have effectuated elsewhere.

**RC:**   *Figures 2 and 3: these figures did not render properly in the PDF I downloaded, using two different widely-used PDF viewers on a Mac. Given that, I can't properly assess these figures.*

AR:   We are sorry for this. It turns out that there were embedded fonts in the PDFs that we uploaded. During the processing by NHESS the figures were not rendered properly and not checked. To make sure, that the figures are rendered properly, we will use the PNG-format now. You can find the figures at the end of this document.

**2.   Comments and responses to the Referee 2**

**RC:**   *L.23: Please define the abbreviation HPE, preferably at the first occurrence of the full name.*

AR:   Thank you for spotting this. We added the abbreviation now at the first occurrence in line 23:

> In the context of flood hazard assessment, one option for counterfactual scenario design is to spatially shift the location of a heavy precipitation event (HPE) in order to assess the impact that it could have effectuated elsewhere.

**RC:** *L.52: Formatting error in „HQextrem"*

AR: Fixed. It is now: $HQ_\text{extreme}$.

**RC:** *L.75: The SCS-CN method is often criticized for its simplicity and its empirical character. Could you justify why this method is appropriate for calculating infiltration and surface runoff in your study?*

AR: We chose the SCS-CN method because it is widely known and accepted and requires only a few parameters. All flash flood models that we are aware of use the same method for the calculation of the effective precipitation. E.g. the models used by in Borga et al. (2007), Marchi et al. (2010), Ruiz-Villanueva et al. (2012), Tarolli et al. (2013), Gaume et al. (2004), Versini et al. (2010) and Emmanuel et al. (2017).

Due to the restriction of the number of references, we do not cite all of these references in the brief communication. Further justification of the model approach can, however, be found in Voit and Heistermann (2024).

Apart from that, the application of more advanced models for runoff generation is typically limited by the robust parameterisation, specifically in small basins.

Altogether, we suggest to enhance section 2.3 after l. 82 of the preprint:

> First, the effective rainfall is estimated using the SCS- CN method (U.S. Department of Agriculture-Soil Conservation Service, 1972). The SCS- CN method is widely used in flash flood modelling while more advanced modelling approaches are typically difficult to parameterize specifically in small catchments.

**RC:** *L.79: More information about the hydrologic model is needed. Which hydrological model is used? If an own model is used, please mention this.*

AR: You are right, this is unclear. We developed our own implementation of the model for the study in Voit and Heistermann (2024) which is, however, very similar to the modelling approaches used on other flash flood studies. We will change the first sentence of section 2.4 for clarification:

> We specifically tailored the hydrological model to represent flash flood events in small- to medium-sized basins. A comprehensive model description can be found in Voit and Heistermann (2024).

**RC:** *L.91: "We model the quick runoff" is a too short for the methods. Could you elaborate more how you have modelled quick runoff?*

AR: Quick runoff here corresponds to the effective rainfall as obtained from the SCS-CN method. This should become clear from section 2.4. In order to clarify this, we supplement the sentence in ll. 82-82 as follows:

> Secondly, the geomorphological instantaneous unit hydrograph (GIUH), as derived from the DEM, is used to represent the concentration of quick runoff (i.e. of the effective rainfall).

**RC:** *L.93: In FFA studies, at least 30 years are usually used to calculate the annual maxima series and to generate the distribution function. Please explain in more detail why 23 years are sufficient in your analysis.*

**AR:** This is a major limitation of using RADKLIM data for extreme value statistics, and we acknowledge it transparently in line 94. However, the answer on whether a time series is long enough depends on the return period you want to address by FFA. To our knowledge, an extrapolation up to 3 times the length of the time series is often considered as justified. For this reason we wrote:

> Given the length of our yearly maxima series (2001-2022), we consider the estimation of the 50-year return level as reasonably robust, while the 200-year return level will obviously be highly uncertain.

**RC:** *Fig. 2: Description of X-axis and Y-axis is missing &*

Fig. 2: What is the meaning of "0 50" and so on? Even if it is explained in the caption, the subplot titles should be more specific. &

Fig. 2: And add an explanation for two lines (green, blue) to the legend. At the moment, only one line (red) is explained.

**AR:** Unfortunately, the figure is incomplete. We submitted it in PDF-format and it must have been compiled faulty. You can find the correct figure (Fig. 1) at the end of this document.

**RC:** *Fig. 3: Description of X-axis and Y-axis is missing. &*

Fig. 3: What does "0 50" and so on mean? Even if it is explained in the caption, the subplot titles should be more precise. &

Fig. 3: The legend needs to be reformatted. The lines are partly covered by the legend. The green line has no explanation. The explanation for the other two lines (red, blue) should be presented consistently, either on the left or the right-side. &

Fig. 3: The legend needs to be reformatted. The lines are partly covered by the legend. The green line has no explanation. The explanation for the other two lines (red, blue) should be presented consistently, either on the left or the right-side

**AR:** Also this figure was not rendered correctly when the preprint was generated by Copernicus. You can find the correct figure (Fig. 2) at the end of this document.

**RC:** *Fig. 3: Why m³/s/km² 0 6 and Why 2 0 6?*

**AR:** Also here the unit got formatted wrongly after we had submitted the figure as PDF. The unit of the unit peak discharge (UPD) should be: $m/s/(km)^{0.6}$.

While some studies use no exponent for the catchment area, other studies suggest to limit the influence of the upstream area by using an exponent to account for the decrease of unit discharges with upstream catchment areas. Figure 3 at the end of this document illustrates this.

For instance, the upper limit for Flash floods for Europe (envelope curve) is evaluated as $Q = 100 * A^{0.6}$ in Gaume et al. (2008). For this reason we also decided to use 0.6 as exponent for the catchment area.

**RC:** *Results and discussion: Could you perhaps add sub-chapters to make it easier to read?*

AR: We suggest to split this section into two subsections. The first sub-section about the comparison between local and global counterfactuals and the second one about the case study regarding the Danube flood in June 2024.

**References**

Borga, M., Boscolo, P., Zanon, F., and Sangati, M.: Hydrometeorological analysis of the 29 August 2003 flash flood in the Eastern Italian Alps, Journal of hydrometeorology, 8, 1049–1067, 10.5194/nhess-2023-22410.1175/JHM593.1, 2007.

Emmanuel, I., Payrastre, O., Andrieu, H., and Zuber, F.: A method for assessing the influence of rainfall spatial variability on hydrograph modeling. First case study in the Cevennes Region, southern France, Journal of Hydrology, 555, 314–322, 10.5194/nhess-2023-22410.1016/j.jhydrol.2017.10.011, 2017.

Fontaine, T. A. and Potter, K. W.: Estimating probabilities of extreme rainfalls, Journal of Hydraulic Engineering, 115, 1562–1575, 10.5194/nhess-2023-22410.1061/(ASCE)0733-9429(1989)115:11(1562), 1989.

Gaume, E., Livet, M., Desbordes, M., and Villeneuve, J.-P.: Hydrological analysis of the river Aude, France, flash flood on 12 and 13 November 1999, Journal of hydrology, 286, 135–154, 10.5194/nhess-2023-22410.1016/j.jhydrol.2003.09.015, 2004.

Gaume, E., Bain, V., Bernardara, P., Newinger, O., Barbuc, M., Bateman, A., Blaškovičová, L., Blöschl, G., Borga, M., Dumitrescu, A., et al.: A compilation of data on European flash floods, Journal of Hydrology, 367, 70–78, 10.5194/nhess-2023-22410.1016/j.jhydrol.2008.12.028, 2008.

Hansen, E. M.: Probable maximum precipitation for design floods in the United States, Journal of Hydrology, 96, 267–278, 10.5194/nhess-2023-22410.1016/0022-1694(87)90158-2, 1987.

Lompi, M., Caporali, E., Mediero, L., and Mazzanti, B.: Improving flash flood risk assessment using a simple approach for extreme rainfall scaling and storms transposition, Journal of Flood Risk Management, 15, e12 796, 10.5194/nhess-2023-224https://doi.org/10.1111/jfr3.12796, 2022.

Marchi, L., Borga, M., Preciso, E., and Gaume, E.: Characterisation of selected extreme flash floods in Europe and implications for flood risk management, Journal of Hydrology, 394, 118–133, 10.5194/nhess-2023-22410.1016/j.jhydrol.2010.07.017, 2010.

Merz, B., Nguyen, V. D., Guse, B., Han, L., Guan, X., Rakovec, O., Samaniego, L., Ahrens, B., and Vorogushyn, S.: Spatial counterfactuals to explore disastrous flooding, Environmental Research Letters, 10.5194/nhess-2023-22410.1088/1748-9326/ad22b9, 2024.

Montanari, A., Merz, B., and Blöschl, G.: HESS Opinions: The Sword of Damocles of the Impossible Flood, EGUsphere, 2023, 1–20, 10.5194/nhess-2023-22410.5194/egusphere-2023-2420, 2023.

Ruiz-Villanueva, V., Borga, M., Zoccatelli, D., Marchi, L., Gaume, E., and Ehret, U.: Extreme flood response to short-duration convective rainfall in South-West Germany, Hydrology and Earth System Sciences, 16, 1543–1559, 10.5194/nhess-2023-22410.5194/hess-16-1543-2012, 2012.

Tarolli, M., Borga, M., Zoccatelli, D., Bernhofer, C., Jatho, N., and Janabi, F. a.: Rainfall space-time organization and orographic control on flash flood response: the Weisseritz event of August 13, 2002, Journal of Hydrologic Engineering, 18, 183–193, 10.5194/nhess-2023-22410.1061/(ASCE)HE.1943-5584.0000569, 2013.

Versini, P.-A., Gaume, E., and Andrieu, H.: Application of a distributed hydrological model to the design of a road inundation warning system for flash flood prone areas, Natural Hazards and Earth System Sciences, 10, 805–817, 10.5194/nhess-2023-22410.5194/nhess-10-805-2010, 2010.

Voit, P. and Heistermann, M.: A downward-counterfactual analysis of flash floods in Germany, Natural Hazards and Earth System Sciences Discussions, 2024, 1–23, 10.5194/nhess-2023-22410.5194/nhess-2023-224, 2024.

Vorogushyn, S., Han, L., Apel, H., Nguyen, V. D., Guse, B., Guan, X., Rakovec, O., Najafi, H., Samaniego, L., and Merz, B.: It could have been much worse: spatial counterfactuals of the July 2021 flood in the Ahr valley, Germany, Natural Hazards and Earth System Sciences Discussions, 2024, 1–39, 10.5194/nhess-2023-22410.5194/nhess-2024-97, 2024.

Wright, D. B., Smith, J. A., and Baeck, M. L.: Flood frequency analysis using radar rainfall fields and stochastic storm transposition, Water Resources Research, 50, 1592–1615, 2014.

Wright, D. B., Yu, G., and England, J. F.: Six decades of rainfall and flood frequency analysis using stochastic storm transposition: Review, progress, and prospects, Journal of Hydrology, 585, 10.5194/nhess-2023-22410.1016/j.jhydrol.2020.124816, 2020.

Zhou, Z., Smith, J. A., Wright, D. B., Baeck, M. L., Yang, L., and Liu, S.: Storm Catalog-Based Analysis of Rainfall Heterogeneity and Frequency in a Complex Terrain, Water Resources Research, 55, 1871–1889, 10.5194/nhess-2023-224https://doi.org/10.1029/2018WR023567, 2019.

[Figure]

Figure 1: The cumulative density distributions show, for different subbasin sizes [a) < 50 km², b) 50-200 km², c) 200-750 km²], the ratio between three different discharge estimates and the 50-year return level: (1) the local counterfactual peak discharge (green), (2) the global counterfactual peak discharge (purple), (3) the 200-year return level (orange).

[Figure]

Figure 2: Case study of the recent heavy precipitation event from May 30 to June 4, 2024: the black lines show the simulated unit peak discharge (UPD) of the event for all subbasins within the Danube catchment with a return period > 20 years; for comparison, the green lines show the local counterfactual UPD and the orange lines the 200-year return level estimated from simulations between 2001 and 2022.

[Figure]

Figure 3: Unit peak discharge (UPD) for a discharge of 100 $m^3/s$ and changing catchment sizes. Different exponents are used to limit the influence of the catchment area on the UPD: no exponent (blue), exponent=0.6 (orange), exponent=0.8 (green).